# Hypothermically Stored Adipose-Derived Mesenchymal Stromal Cell Alginate Bandages Facilitate Use of Paracrine Molecules for Corneal Wound Healing

**DOI:** 10.3390/ijms21165849

**Published:** 2020-08-14

**Authors:** Olla Al-Jaibaji, Stephen Swioklo, Alex Shortt, Francisco C. Figueiredo, Che J. Connon

**Affiliations:** 1Biosciences Institute, Newcastle University, International Centre for Life, Central Parkway, Newcastle upon Tyne NE1 3BZ, UK; O.M.H.Al-Jaibaji2@newcastle.ac.uk (O.A.-J.); steve.swioklo@atelerix.co.uk (S.S.); francisco.figueiredo@newcastle.ac.uk (F.C.F.); 2Atelerix Ltd., The Biosphere, Newcastle upon Tyne NE4 5BX, UK; 3UCL Institute of Ophthalmology, London EC1V 9EL, UK; a.shortt@ucl.ac.uk; 4Department of Ophthalmology, Royal Victoria Infirmary & Newcastle University, Newcastle upon Tyne NE1 4LP, UK

**Keywords:** cornea, stem cells bandage, paracrine factors, adipose-derived stem cells, hypothermic storage, sodium alginate, wound model

## Abstract

Adipose-derived mesenchymal stromal cells (Ad-MSCs) may alleviate corneal injury through the secretion of therapeutic factors delivered at the injury site. We aimed to investigate the therapeutic factors secreted from hypothermically stored, alginate-encapsulated Ad-MSCs’ bandages in in vitro and in vivo corneal wounds. Ad-MSCs were encapsulated in 1.2% *w*/*v* alginate gels to form bandages and stored at 15 °C for 72 h before assessing cell viability and co-culture with corneal scratch wounds. Genes of interest, including HGF, TSG-6, and IGF were identified by qPCR and a human cytokine array kit used to profile the therapeutic factors secreted. In vivo, bandages were applied to adult male mice corneas following epithelial debridement. Bandages were shown to maintain Ad-MSCs viability during storage and able to indirectly improve corneal wound healing in vivo. Soluble protein concentration and paracrine factors such as TSG-6, HGF, IL-8, and MCP-1 release were greatest following hypothermic storage. In vivo, Ad-MSCs bandages-treated groups reduced immune cell infiltration when compared to untreated groups. In conclusion, bandages were shown to maintain Ad-MSCs ability to produce a cocktail of key therapeutic factors following storage and that these soluble factors can improve in vitro and in vivo corneal wound healing.

## 1. Introduction 

Corneal healing following injury requires the activation of complex cytokine cascades involving the epithelial cells, stromal cells, tear film, immune cells, nerves, and lacrimal glands [1,2]. Wound recovery involves apoptosis of injured cells, mediated by the release of interleukin-1 (IL-1) and tumour necrosis factor (TNF) by the epithelial cells [3]. Corneal stromal cells (CSC) are also triggered to differentiate into an active form called fibroblasts and myofibroblasts that are associated with increased production of hepatocyte growth factor (HGF) and keratinocyte growth factor (KGF) [4]. HGF and KGF are important in the regulation of epithelial cells proliferation, migration, and differentiation. IL-1 is also known to regulate important remodelling enzymes such as collagenase and metalloproteinase, as well as chemokines (IL-8 and monocyte chemotactic protein MCP-1). Platelet-derived growth factor (PDGF), expressed by epithelial cells, is released following injury into the stroma to modulate corneal fibroblasts chemotaxis and proliferation. CSC at the injury site are known to undergo apoptosis, and lacrimal glands produce HGF and epidermal growth factor (EGF) that aid in regulating proliferation, migration, and differentiation until fibroblasts and myofibroblasts can repopulate the stroma and restore homeostasis [5]. Thus, damage to the cornea can lead to wound healing disruption. While corneal transplant is the most widely used surgical procedure to restore sight in patients with corneal damage, a study suggested that 12.7 million people around the world are on the waiting list for such transplants [6]. Thus, new alternatives to corneal stromal repair are sought. One such way is to use a cellular therapy, in which the cells can respond to the wound environment and subsequently act therapeutically via the release of paracrine factors. 

Adipose-derived mesenchymal stromal cells (Ad-MSCs) have previously shown a significant effect on wound healing and regenerative medicine [7,8,9]. They are easily isolated and maintained in vitro, able to differentiate into multilineages, highly proliferative, able to produce a cocktail of paracrine factors, and immune-privileged [10]. In therapy, Ad-MSCs can be either used directly in the target tissue or used as a delivery vehicle of bioactive factors [11]. Ad-MSCs are known to produce a cocktail of paracrine factors such as HGF, IGF, vascular endothelial growth factor (VEGF), KGF, and transforming growth factor beta (TGF-ß) that have proliferative and regenerative effects in response to wounds [12,13]. They can also produce anti-inflammatory cytokines such as IL-1a, IL-6, IL-8, IFNy, MIP2, and MCP-1 [14,15]. In addition, they have angiogenic potential effected by growth factors such as VEGF [16,17]. 

In corneal injuries, Ad-MSCs’ ability to regenerate the damaged cornea has been validated in multiple clinical and animal studies [18,19,20,21]. These studies provide evidence that Ad-MSCs, following injection to the site of injury are able to reduce inflammation while improving cell function via a paracrine activity [22,23]. However, low cell engraftment and low cell survivability have been found at the wound site [24]. The introduction of biomaterials into cell therapy has the potential to overcome cell transplantation issues by ensuring localisation, maintenance, viability, and preservation of cells function whilst protecting cells from external harm [25]. Cell transportation is often overlooked but nonetheless is an acute logistical step before transplantation. Transporting live cells involves maintaining cell bioactivity without losing their viability and function [26,27]. Alginate gel, a natural hydrogel derived from seaweed, has previously been used to encapsulate Ad-MSCs for a period of days to weeks across a range of normothermic and hypothermic temperatures [28,29]. It has been found that alginate gels are able to maintain cell viability and function whilst allowing cells to exchange nutrients [30,31]. Studies also found that alginate gels are able to protect cells during hypothermic storage via membrane stabilisation and provide protection against mechanical stress, but osmotic shock that can also otherwise affect cells during storage and recovery [32]. 

Thus, we sought to combine alginate gels ability in storing cells with their ability to allow the release of growth factors to form ‘stem cell bandages’. To our knowledge, this is the first time that the combined effect of alginate gels, storage, stem cells, and corneal wound healing has been shown. Thus, this investigation may highly benefit our management of corneal injury in the near future. In this study, we explore stem cell bandages comprised of Ad-MSCs their maintenance during short-term storage and subsequent efficacy in wound repair. The ability for Ad-MSC bandages to release paracrine factors was investigated in vitro, in relation to CSC scratch-wound recovery, soluble proteins, and collagen production. Key growth factors were tested for both CSCs and Ad-MSCs. In vivo, Ad-MSCs bandages were placed onto the surface of wounded mouse corneas and assessed with histology studies. Taken together, these data provide compelling evidence of alginate gels ability to maintain Ad-MSCs viability and function during storage, in addition to the release of paracrine factors during storage (IGF, VEGF, αSMA, and HIF1A) and following scratch healing (TSG6, HGF, REX1, IL-6, IL-8, VEGF, and MCP-1). Ad-MSCs bandages were also shown to maintain specific stem cell marker expression, and their ability to reduce inflammation in a wounded mouse model whilst improving tissue recovery. 

## 2. Results

### 2.1. Ad-MSC Bandages Maintain Ad-MSCs Viability during Hypothermic Storage 

Effective cell therapy requires the cells to be active and functional following storage and transportation. Ad-MSCs cell recovery was tested following 72 h storage at 15 °C. Previously, we tested a range of temperatures, comparing viable recovery with non-encapsulated controls and found that 15°C to be the optimum temperature that maintains cells viable recovery in both control and encapsulated samples [29]. 

Herein, Ad-MSCs were encapsulated in 1.2% alginate to form a bandage and stored at 15 °C for 72 h comparing viable recovery with non-encapsulated controls. We found that storage achieved a viable cell recovery of 77% ± 2.9%, a significantly higher recovery when compared to control samples 54.6% ± 4.1% (*p* = 0.0463). This indicated that alginate did indeed protect cell viability during storage (Figure 1A) in line with the product description (Atelerix, Newcastle Upon Tyne, UK). This storage is agreeable with the FDA’s minimum acceptable cell viability of 70% for cellular products [33]. Cell morphology was examined upon return to culture and attachment to tissue plastic, where they showed a normal spindle-shaped fibroblastic-like morphology, indistinguishable phenotype to the normally grown cell cultures (Figure 1B). This indicated that Ad-MSCs proliferation potential was unaffected by storage. Ad-MSC phenotypic surface markers were assessed following storage to determine whether storage affected cell marker expression using flow cytometry compared with the literature. The expressions of positive CD90 and CD73 and of negative CD45 and CD14 markers were maintained following storage and of HLA-ABC and HLA-DR remained unchanged (Figure 1C), suggesting that alginate maintained cells immunophenotypic expression. 

### 2.2. Ad-MSCs Bandages Improve Wound Closure via Release of Paracrine Factors 

Ad-MSC bandages were stored at 15 °C for 72 h before incubating with a co-culture of CSC (Figure 2A). Compared to no-cell bandage controls, the presence of 15 °C-stored Ad-MSC showed an improved scratch area coverage of the CSC at times 20, 30, and 40 h post-scratch (*p* = 0.001, <0.0001 and <0.0001) and when compared to SFM controls (*p* = 0.0021, 0.0006 and <0.0001) (Figure 2B), whereas no differences were found between stored and non-stored alginate bandages. Ad-MSC bandages stored at 15 °C also showed a significant increase in CSC number following scratch healing when compared to no-cell bandage and SFM controls (*p* = 0.0497 and 0.0378) (Figure 2C). In addition, Ad-MSCs’ viable recovery following healing averaged to 73.9% (data not shown), indicating no significant loss in cell number following culture with CSC. This suggests that 15 °C-stored Ad-MSC within alginate improved cell area coverage via the release of paracrine factors that induced CSC to proliferate. 

### 2.3. 15 °C Storage of Ad-MSCs Bandages Effected an Increase in Soluble Protein Concentration

Bradford assay was performed using the media produced following co-culture of scratch wound closure to determine soluble protein content in that solution (Figure 3A). Following wound healing, the soluble protein present in the media from previously stored Ad-MSCs bandage measured to be 19 ± 4.0 mg/µL. Although no significant differences were found between the control samples and Ad-MSCs bandages, a slight increase in protein production was noticed in the 15 °C-stored bandages. This increase suggests that Ad-MSCs paracrine production is enriched following storage at 15 °C. Sirius red staining was performed on CSC cultures to measure insoluble collagen production, where a small increase was found in the stored Ad-MSC condition (Figure 3B). This slight increase in CSC protein and collagen production seen can be attributed to the increase in CSC proliferation (Figure 3C). However, no differences were found upon normalising collagen amount to CSC number (Figure 3D). 

### 2.4. Corneal Stromal Cells Maintain Their Molecular Marker Expression following Wound Healing

CSC molecular phenotype marker expression was analysed at the gene transcription level following their exposure to Ad-MSCs bandages. The relative mRNA expression levels of decorin (*DCN*), lumican (*LUM*), vimentin (*VIM*), alpha smooth muscle actin (*αSMA*), and aldehyde dehydrogenases 3A1 (*ALDH3A1*) were assessed by qPCR (Figure 4). The transcription of corneal proteoglycan genes coding for *DCN* was downregulated in the 15 °C stored bandages when compared to SFM control (*p* = 0.0182) (Figure 4A). No significant differences were found in *LUM* (Figure 4B)*,* corneal structural protein-coding for *VIM* (Figure 4C)*,* and myofibroblast marker coding for αSMA (Figure 4D) expressions. However, corneal crystalline marker ALDH3A1 was upregulated following exposure to 15 °C-stored bandages compared to SFM control, no-cell bandages, and non-stored bandage (*p* = 0.0040, 0.0022, and 0.0047, respectively) (Figure 4E). Thus, storage only significantly affected ALDH3A1 compared to non-stored samples, and an increase in crystallin expression is a positive finding for potentially maintaining corneal transparency. 

### 2.5. 15 °C-Stored Ad-MSC Bandages Increase Expression of Key Genes following Scratch Healing 

Ad-MSCs markers were measured at the gene transcription level comparing Ad-MSC bandages after storage to bandages after healing (Figure 5). Results suggest an upregulation of anti-inflammatory gene coding tumour necrosis factor-inducible gene 6 (TSG-6) and HGF during the healing process, but downregulation of VEGF, hypoxia-inducible factor 1A (HIFA1), ASMA and IGF (*p* = 0.0374, 0.0172 and 0.0018) (Figure 5A–F). Ad-MSCs were also found to maintain their stem cell phenotype via expressing high levels of REX1 during healing (*p* = 0.0479) (Figure 5G). These results suggest that Ad-MSCs have an active role in wound healing and that these cells may be affected by CSC presence. 

### 2.6. Hypothermically Stored Ad-MSCs Bandages Produce a Cocktail of Paracrine Factors in Response to the Environmental Cues 

Human XL Cytokine Array kit was used to detect the factors within the media formed following scratch healing comparing SFM, no-cell bandages, non-stored, and 15 °C-stored Ad-MSC bandages (Figure 6). The resulting relative amounts of different proteins were plotted in a radar graph where proteins were shown to be upregulated in the 15 °C storage condition, such as epithelial-neutrophil-activating peptide (CXCL5/ENA-78), growth-regulated alpha protein (CXCL1/GROa), interleukin (IL-6, IL-8, IL-17A), microphage migration inhibitory factor (MIF), pentraxin3, monocyte chemoattractant protein (CCL2/MCP-1, CCL7/MCP-3), dipeptidyl peptidase-4 (CD26/DPPIV), growth/differentiation factor 15 (GDF15), angiogenin, endoglin, thrombospondin1, and VEGF. This indicates that Ad-MSCs produce paracrine factors in response to the scratch wound. 

### 2.7. In Vivo Study: Hypothermically Stored Ad-MSC Bandages Reduced Immune Infiltration while Improving Wound Healing

To verify the effect of 15 °C stored Ad-MSC bandages on corneal healing in vivo, epithelial debrided mouse corneas were treated with alginate bandages with or without Ad-MSCs. Ad-MSCs formed in an approx. 1 mm thickness gel at a density of 20 × 10^6^ cells/mL and stored for 3 days at 15 °C. Following injury to the cornea, 3.5 mm diameter gels either without cells or with cells were applied directly to the wounded cornea, and the eyelids were sutured closed (Figure 7A). Bandages were left in situ for 4 days, following which the suture and bandage were removed and the eye evaluated. Images of the corneas before enucleation showed a qualitative reduction in corneal haze in wounds treated with Ad-MSC bandages when compared to cell-free bandages (Figure 7B). H&E staining and immunofluorescence showed low levels of immune cells in the normal and Ad-MSC bandage-treated corneas but high levels of monocytes and neutrophils (Figure 7C) in the epithelial debrided untreated corneas. αSMA expression by keratocytes was absent in the normal and Ad-MSC bandage-treated injured corneas but present in the wounded untreated corneas. These data indicated that soluble factors from the Ad-MSCs bandages were successfully passed from the therapeutic bandage to the wound bed and were effective in reducing corneal inflammation and haze. 

## 3. Discussion 

Damage to the cornea is usually caused by trauma, inflammation, pathological changes, or systemic diseases [34]. The severity of the damage can lead to permanent corneal vascularisation, scarring, conjunctivalisation, opacification, and keratinization. Current treatments often include corneal transplantation if the visual axis is affected, which has limitations due to donor availability and compatibility. To restore corneal transparency and integrity, Ad-MSCs therapy has been investigated [35,36,37,38,39]. Using clinical and animal studies, Ad-MSCs therapy has recently proven to be a promising method used in corneal reconstruction with multiple treatment approaches [40,41,42,43]. Moreover, it has been found that Ad-MSCs are able to provide paracrine factors in response to the wound but also able to avoid host immune rejection and transdifferentiate into multilineages [10,17,44]. 

Previously, we showed that alginate gels are suitable for storage and transport of immobilised embryonic stem cells, limbal epithelial stem cells, mesenchymal stem cells, and multipotent adult progenitor cells [28,29,45,46], proving that alginate is able to protect cells phenotype. Comparable with literature, Ad-MSCs bandages stored at 15 °C showed optimum cell recovery and a similar immunophenotype marker expression where they had low expressions of major histocompatibility class II (MHCII) and absence of CD45 and CD14, while expressing CD90 and CD73, thus proving the suitability of our encapsulation method and the functionality of alginate in maintaining the encapsulated cells progenitor phenotype. 

To determine alginate influence on the potential production of paracrine factors from Ad-MSCs, we examined Ad-MSCs’ ability to produce paracrine factors from within the bandage. Ad-MSCs key features are their production of chemokines, cytokine, and growth factors in response to injury [12,47]. Hypothermically stored Ad-MSCs bandages were found to positively affect CSC scratch wound closure without direct contact with the injured cells. Although no significant differences were found in soluble protein concentration and collagen production, a slight increase was noticed in the 15 °C Ad-MSCs bandages. This may be attributed to the increase in CSC number following culture with stored Ad-MSCs bandages. This suggests that cells might be sensing the environmental cues to produce factors. Cells are able to produce specific cues during injury [48]; thus, we expect CSC to produce specific cues, whereas Ad-MSCs respond to these cues by producing factors as an aid to wound healing. Thus, our data have shown a difference in the amount of healing in the presence of Ad-MSCs. It is shown that hypothermic storage is able to preserve cells [29,49]. From our data, we also found that hypothermic storage is able to enhance cells activation, resulting in the production of more factors that aid in wound healing. Although this may be attributed to storage being a hypoxic-induced condition where there is a reduced oxygen circulation. Hypoxia is found to enhance Ad-MSCs’ ability to secrete similar growth factors to those seen here [50,51,52]; this, however, was not tested in our experiment.

Key CSC expression markers were measured at the gene transcription level, coding for corneal proteoglycan genes (DCN and LUM), fibroblasts markers (αSMA and VIM), and corneal crystallin marker ALDH3A1. DCN and LUM, are important corneal keratin sulfate proteoglycans that play a major role in transparency and morphogenesis, whereas VIM and αSMA are myofibroblasts markers [53,54]. ALDH3A1 is believed to be expressed by keratocytes to protect the cornea against UV-induced oxidative stress and is associated with the maintenance of corneal clarity [55,56,57]. Our data suggest DCN, LUM, VIM, and αSMA to be downregulated when exposed to hypothermically stored Ad-MSC bandages, whilst ALDH3A1 was upregulated, whereas the upregulation of LUM and αSMA was noticed following exposure to non-stored Ad-MSC bandages samples and downregulation of ALDH3A1. This suggests that storage enhances Ad-MSCs to produce beneficial growth factors for wound healing following hypothermic storage. On the other hand, alginate is suggested to absorb the proteins and ions within the media; this effect has been suggested when using alginate to encapsulated limbal epithelial stem cells, where authors found alginate to absorb proteins ranging from approx. 35 to 55 kDa to elicit an effect on cell growth [58]. Alginate function in stabilizing cell membranes during storage is due to the absence of the extracellular matrix, thus, resulting in reducing the effect of mechanical stress and osmotic shock during storage and recovery [59]. Moreover, alginate strongly influences the matrix ions, solute diffusion, and water [60,61]. Thus, it may sequester the ions from the media and in turn affect the cells. 

In addition, key Ad-MSCs markers were tested on the gene transcription level following 15 °C storage and healing. The REX1 gene is usually highly expressed by Ad-MSC and is used as Ad-MSCs phenotypic marker [62,63]. Our data showed high expression of REX1 following healing, which suggests that Ad-MSCs maintain their phenotype expression without proliferation or differentiation from within the bandage. Of the soluble factors known to be produced by Ad-MSCs in response to injury, TSG-6 and HGF were upregulated following healing, while hypoxia markers VEGF, HiF1A, and αSMA were upregulated during storage. These results suggest that storage may produce a hypoxic environment for the cells. Although not further tested in our experiments, it is suggested that Ad-MSCs can become highly active during hypoxic conditions [51,64,65]. In line with that, our results showed that cells used following storage resulted in improved scratch wound healing compared to non-stored conditions. 

It is well known that Ad-MSCs produce a variety of inflammatory cytokines and chemokines such as IL-6, IL-8, and MCP-1 [66,67]. These factors have previously been shown to be upregulated during corneal wound healing [68,69]. Specifically, IL-6 and IL-8 were shown to regulate leukocyte infiltration during wound healing, whilst triggering fibroblasts and keratocytes migration [70,71]. MCP-1 is an important chemokine that regulates macrophage recruitment [72]. Our data show a high expression of these proteins in the 15 °C-stored Ad-MSCs compared to the other conditions. Inflammatory regulatory cytokine ENA-78 found to be enhanced in response to IL-1 alpha or TNF-alpha stimulation [73]; here, ENA-78 was upregulated in the 15 °C-stored Ad-MSCs samples, suggesting exposure to hypothermic storage may increase in angiogenic response. Other cytokines were found to be upregulated in the 15 °C stored Ad-MSCs samples, such as pentraxin3 is suggested to bind to apoptotic cells to inhibit their recognition by dendritic cells and regulated their clearance from the injury site while improving wound healing [74,75]. GDF15, a distant member of TGF-ß superfamily, which regulates various cellular processes and is a factor known to be involved in many diseases’ pathobiology [76,77]. DPPIV upregulation is associated with signal transduction, immune regulation and cell apoptosis, and it can suppress the development of fibrosis [78]. Moreover, an antiangiogenic factor, Thrombospondin-1 [79,80], upregulation at 15 °C-stored bandages suggests Ad-MSCs induced antiangiogenic factor production to promote corneal wound healing, thus signifying the beneficial ability of using Ad-MSCs as stem cell bandage. Despite the limited number of biological repeats, our data suggest that Ad-MSCs bandages produce paracrine factors in response to injury, thus supporting our belief on the effect of Ad-MSCs bandages may have on corneal therapy. 

Epithelial scraped mouse models were used to further investigate in vivo the benefits of hypothermically stored Ad-MSC bandages. It has been shown that Ad-MSCs can promote wound healing in various animal models using different approaches [38,81,82]. Our data show a reduction in corneal haze with an improvement in corneal healing shown by H&E staining. Moreover, immunohistochemistry data showed a reduction in immune cell infiltration and αSMA production in the Ad-MSCs group compared to no-cell bandage group. These data (despite the limited number of animals tested) suggest that, within our system, Ad-MSCs were seen to reduce inflammation in the injured eye animal model. Drawing upon our in vitro data, this could be due to HGF production that inhibits myofibroblasts generation and production of disorganised ECM [83]. IGF was also shown to support cell proliferation and enhance wound closure [84,85], whereas TGF-β and IL-8 can be responsible for suppressing inflammation whilst improving healing [86]. Furthermore, ENA-78, GROa, pentraxin3, and GDF15 regulate inflammatory cells infiltration and activation [85]. However, there are limitations of our data as the number of animals used was small, suggesting that future studies with more animals are recommended. 

In the present study, hypothermically stored Ad-MSC bandages stimulated the migration and proliferation of CSC and improved in vitro and in vivo wound healing. Ad-MSCs secreted a variety of cytokines, growth factors, and chemokines, such as HGF, IGF, VEGF, IL-6, IL-8, MCP-1, and TSG-6 in response to injury [52,87,88]. Ad-MSCs are shown to improve healing following storage and improve corneal injury. Evidence has also been provided that these Ad-MSCs release a cocktail of paracrine factors that plays an important role in stimulating wound healing response [89,90,91]. In vivo, alginate-encapsulated Ad-MSCs applied to the ocular surface of an epithelial debrided mouse cornea reduced immune cells infiltration while improving corneal haze. Consistent with previous studies [20,21], our results showed Ad-MSCs bandages can stimulate CSC migration and proliferation while upregulating ECM-released genes and cell survival. In addition, we provide evidence that this was due to soluble factors generated from immobilised cells; moreover, this cocktail may change in response to a wound environment (Figure 8). 

In conclusion, we present the ability to store and maintain Ad-MSCs viability during hypothermic storage via encapsulating them within alginate gel to form a novel stem cell bandage. We have also shown the improved effect of using Ad-MSCs bandages in healing CSC in vitro and in vivo chemical wound. Therefore, the use of alginate to form Ad-MSC bandage present a promising therapeutic approach for the treatment of corneal injuries. 

## 4. Materials and Methods

### 4.1. Human Corneal Stromal Cell Culture

Human corneal stromal cells were obtained from cornea-scleral rings remaining after removal of the central cornea for clinical transplantation supplied by the NHS Blood and Transplant (NHSBT) Cornea Transplantation Service eye bank in Manchester and Bristol, UK. Human tissue was handled according to the tenets of the Declaration of Helsinki, and informed consent was obtained for research use of all human tissue from the next of kin of all deceased donors and had ethical approval (NRES Committee North East-Newcastle and North Tyneside 1 (REC number: 11/NE/0236, protocol number 5466; October 2013)). 

Human corneal rings were debrided of endothelial and epithelial cells before mincing and incubating in a humidified incubator (37 °C, 5% CO_2_) with Dulbecco’s Modified Eagle Medium (DMEM/F12) (ThermoFisher Scientific, Loughborough, UK) supplemented with 5% fetal bovine serum, 1% penicillin-streptomycin (ThermoFisher Scientific, Loughborough, UK) and 2 g/L collagenase type I (Sigma-Aldrich, Irvine, UK). Five hours following incubation, the cells were dissociated with 0.25% Trypsin-EDTA (ThermoFisher Scientific, Loughborough, UK) solution and filtered through a 40 μm EASYstrainer™ (Greiner Bio-One, Stonehouse, UK), followed by 5 min centrifugation at 1500× *g*. The pellets were then re-suspended and expanded with DMEM/F12 medium with media change three times a week. Upon confluency, the cells were dissociated for expansion with the TrypLE^TM^ express enzyme (ThermoFisher Scientific, Loughborough, UK). Cells used for the experiments were up to passage 5. 

### 4.2. Human Adipose-Derived Mesenchymal Stem Cells Culture

Human Ad-MSCs were commercially purchased from Invitrogen (ThermoFisher, Loughborough, UK) at passage 1. The cells were received with a Certificate of Analysis detailing the donor’s information (2 female donors aged 43 and 45 years old and 1 male donor aged 63) and the tests performed to check cell quality. In the lab, the cells were expanded in MesenPRO RS^TM^ Medium (Gibco^TM^, ThermoFisher Scientific, Loughborough, UK) supplemented with MesenPRO RS^TM^ growth supplement, 2% antibiotic-antimycotic (100×), and 1% GlutaMAX (100×) and cryopreserved in liquid nitrogen by supplementing them with 20% DMSO and 40% FBS until required. The cells were expanded and used up to passage 5 for all subsequent experiments. 

### 4.3. Cell Plating, Scratch Assay and Assessment of Proliferation 

CSCs were plated during passages 1–5 in a Multidish Nunclon 6 well plate 132 × 88 mm (ThermoFisher Scientific, Loughborough, UK). Cells were seeded at 3 × 10^5^ cells/well with DMEM/F12 supplemented with 5% FBS and 1% penicillin-streptomycin. The plates were incubated for 24 h at 37 °C and 5% CO_2_ before serum starving the cultures for 72 h by incubating with serum-free DMEM/F12 (SFM) supplemented with 1 × 10^−3^ M L-Ascorbic acid (Sigma-Aldrich, Irvine, UK), 1:100 Insulin-Transferrin-Selenium (ITS-G) (Invitrogen, Inchinnan, UK), and 1% penicillin/streptomycin. This medium has been shown to induce the cells to express a keratocytes characteristic phenotype [92]. SFM was used in all subsequent experiments. 

Scratch assays were used to measure cell behaviour in vitro. A 1 mL (1.15 mm diameter) pipette tip was used to introduce a mechanical wound on cells confluent monolayer. Images were taken using 6-well cytonote IPRASENSE lensless holographic microscope (Cap Alpha, Avenue de l’Europe, 34830 CLAPIERS, France). Time-lapse images of the scratch were taken with 1 h intervals and processed using ImageJ software (v1.49u). 

Cell proliferation was assessed using Alamar Blue assay (ThermoFisher Scientific, Loughborough, UK). Plates were incubated at 37 °C, and fluorescence was measured at 560 nm excitation and 585 mm emission using Varioskan^TM^ LUX Multimode Microplate Reader (ThermoFisher Scientific, Loughborough, UK). 

### 4.4. Ad-MSCs Bandage Formation and Assessment 

Upon reaching confluence, Ad-MSCs were counted using Live/Dead^®^ staining to determine the required cell number for encapsulation. In 1 mL sodium alginate, 2 × 10^6^ viable cells were encapsulated using a BeadReady™ kit (Atelerix, Newcastle Upon Tyne, UK) with a modified protocol. Briefly, 2× concentrated sodium alginate +/− cells were mixed with SFM before casting in ring-shaped filter papers soaked with the supplied gelation buffer for initial gelation of 2 min followed by 6 min incubation in gelation buffer at room temperature as described previously [45]. The formed alginate rings +/− cells were then stored at 15 °C for 72 h before use in in vitro or in vivo wound healing studies. 

To release the cells for further testing, the Ad-MSCs bandages were dissolved via incubating with the release buffer (Atelerix, Newcastle Upon Tyne, UK) at room temperature with gentle agitation over 3 min followed by centrifugation (1500× *g* for 5 min) and resuspension with media before counting using Live/Dead stain and Countess II FL Automated Cell Counter (ThermoFisher Scientific, Loughborough, UK). 

Following cell release, Ad-MSCs were plated for 24 h in a 24-well plate (Greiner Bio-One, Stonehouse, UK) with MesenPro RS^TM^ medium. Images were taken using light microscopy (Leica Microsystems, Wetzlar, Germany) at 10× magnification. Cell phenotype was compared to normally grown cultures. 

Flow cytometry was used to assess Ad-MSC immune-surface markers following storage and compared to cells grown at normal culture conditions (37 °C and 5% CO_2_). Cells were labelled with fluorescent antibodies CD73, CD90, and HLA-ABC as positive markers, while CD14, CD45, and HLA-DR were labelled as negative markers and IgG2a PE, IgG1 FITC, and IgG1 PE as isotope markers (BD Biosciences, Becton Dickinson, Franklin Lakes, NJ, USA). Twenty minutes following incubation, the tubes were centrifuged and resuspended with 0.1 mL PBS before filtering the solution to remove clumps or cell aggregation using 20 µm filters into a special FACS tube (Flow Cytometry Core Facility, Newcastle University, Newcastle Upon Tyne, UK). An LSRFortessa™ (BD Biosciences, Wokingham, Berkshire, UK) was used to detect cell markers underflow, and the results were plotted using BD FACSDIVA software.

### 4.5. Total Protein Analyses 

Quick Start™ Bradford protein assay (Bio-Rad Laboratories Ltd., Watford, Hertfordshire, UK) was used to measure soluble protein concentration in the media samples following scratch wound healing. Prior to mixing with the dye, the media were concentrated by adding 4× volume ice-cold 70% ethanol (absolute ethanol, VWR, Lutterworth, Leicestershire, UK) before incubating overnight at −20°C. The tubes were then centrifuged at 10,000× *g* at 4 °C, the pellets were allowed to dry at room temperature prior to resuspending with PBS. Bradford assay was used according to the manufacturers’ procedure, 250 µL 1× dye was incubated with 5 µL concentrated protein for 20 min before reading absorbance at 595 nm using VariosKan LUX Multimode Microplate Reader (ThermoFisher Scientific, Loughborough, UK). A standard curve made of bovine serum albumin was used to determine sample concentration. 

### 4.6. Assessment of Collagen Amount 

Picro-Sirius Red (Sigma-Aldrich, Irvine, UK) was used to measure collagen amount produced by the CSCs at the end of each experiment. The stain was used as described previously [93]; briefly, the plates were gently washed using PBS before fixing the cells with 70% ice-cold ethanol (VWR, Soulbury, UK) for 10 min, and the plates were incubated with Sirius red stain overnight at 4 °C with gentle agitation. Plates were then washed and imaged using light microscopy before incubating at room temperature with 1M sodium hydroxide (ThermoFisher Scientific, Loughborough, UK). Dye absorbance was measured at 490 nm, and results were plotted against a standard curve made from Rat Tail Collagen I (First Link Ltd., UK). 

### 4.7. Real-Time PCR

RNA was extracted using RNeasy^®^ Mini Kit (QIAGEN, Manchester, UK) according to the manufacturer instructions. RNA purity and concentration were measured using NanoDrop 2000 spectrophotometer (ThermoFisher Scientific, Loughborough, UK). cDNA synthesis was prepared using RT^2^ First Strand Kit (QIAGEN, Manchester, UK) according to the manufacturer protocol. Real-time PCR reactions were performed in a 48-well Eco™ PCR reaction plate (Illumina, San Diego, CA, USA). Seven microlitres of Syber Green Mastermix (ThermoFisher Scientific, Loughborough, UK) was mixed with 5 µL cDNA sample and 2 µL forward and reverse primer mix. The plates were sealed using Eco ™adhesive seal (Illumina, San Diego, CA, USA) and centrifuged briefly to ensure no bubbles were present. Plates were then transferred to Eco™ Real-Time PCR System (Illumina, San Diego, CA, USA) and incubated at 95 °C for 10 min, followed by 40× cycles of 10 s at 95 °C, 30 s at 60 °C, and 15 s at 72 °C. The primer sequences described in Table 1. 

### 4.8. Proteome Profiler Human XL Cytokine Array 

Human cytokine array kit (R&D Systems, Abingdon Science Park, Abingdon, UK) was used to detect cytokines present in the sample. The kit was used according to the manufacturer’s instructions to give a relative expression of 105 different cytokines. The kit was used with CM taken from Ad-MSCs bandages stored at 15 °C and non-stored following incubation with CSC scratch wound. The resulting membranes were imaged using Amersham™ Imager 600 (GE Healthcare Life Sciences, Buckinghamshire, UK), and ImageJ was used to measure pixel intensity of each captured antibody spot. 

### 4.9. Effect of Stem Cell Bandage Delivery to the Ocular Surface of Wounded Mice

In vivo studies were conducted in compliance with Home Office legislation and the relevant regulatory requirements. Studies were approved by the local ethics committee (Biological Research Unit, UCL Institute of Immunity and Transplantation, Royal Free Hospital) and met NC3R standards. Animal care and handling were in line with ARVO Statement for Use of Animals in Ophthalmic and Vision Research. Adult male NSG (NOD/SCID/Gamma) mice aged 8 to 12 weeks were divided into two groups: those receiving an alginate gel with no cells, and those receiving an alginate gel containing 20 × 10^6^ cells/mL (approximately 60,000 cells per treatment) (*n* = 4). Human Ad-MSCs were formed in 3.5 diameter alginate stem cell bandages, stored, and transported under temperature-monitored condition set at 15 °C for 3 days. Under combined topical and general anaesthesia, the left cornea and limbus were treated with 20% ethanol for 3 min before debridement of the corneal and limbal epithelium, followed by copious irrigation with PBS. n 8-0 proline tarsorrhaphy suture was pre-placed; then, a 3.5 mm diameter gel was placed over the cornea and limbus, the lids were shut, and the suture was tied. The suture was removed after 4 days and the lids opened. Seven days following injury, the eyes were imaged (Leica MZ10 F Stereomicroscope). The animals were sacrificed under terminal anaesthesia, and the eyes were enucleated, fixed with 4% w/v paraformaldehyde, and mounted in OCT. Fifteen-micrometre sections were prepared before Hematoxylin & Eosin and immunohistochemistry staining were carried out. 

### 4.10. Haematoxylin & Eosin Staining

Haematoxylin & Eosin (H&E) were performed as described by Fischer et al. [94]. Tissue sections were washed with 95% and 70% ethanol for 2 min before staining with H stain (Sigma-Aldrich, Irvine, UK) for 2.5 min, followed by 2 dips in acid alcohol differentiation (1% HCL and 70% ethanol) solution and 2 min wash with ammonia water (0.2% ammonium hydroxide in distilled water) blueing solution. The slides were then washed with 95% ethanol before adding Eosin Y solution (Sigma-Aldrich, Irvine, UK) counterstain for 60 s. The section slides were then washed with 95% and 100% ethanol for 5 min, cleared, and mounted with DPx mountant solution for histology (Sigma-Aldrich, Irvine, UK) to preserve the stain. Nikon ECLIPSE TS100 (Nikon Instruments Europe BV, Tripolis100, Amsterdam, Netherlands) inverted microscopy was used to take images using 20×, 10× and 5× magnification using ProgRes^®^ CapturePro (V2.8.8) (Jenoptik, Jena, state of Thuringia, Germany) image analysis software. 

### 4.11. Immunohistochemistry 

R&D system (Abingdon Science Park, Abingdon, UK) protocols were used to stain the tissue sections; briefly, the sections were incubated for 30 min with blocking/incubation buffer containing 2% BSA (First link Ltd., Wolverhampton, UK) and 0.1% Triton^®^ X-100 (ThermoFisher Scientific, Loughborough, UK) in PBS before incubating overnight at 4 °C with primary antibody. Neutrophil antibody (NIMP-R14)—rat anti-mouse (Abcam, Cambridge, UK) diluted in incubation buffer (1:50). Samples were then washed 3× for 15 min before incubating with IgG anti-rate secondary antibody (1:1000 dilution) (Vector Laboratories, Peterborough, UK) and Hoescht 33342 solution (1:2000 dilution) (ThermoFisher Scientific, Loughborough, UK) at room temperature. Slides were then washed and mounted using Prolong^®^ Diamond antifade mountant molecular probe (ThermoFisher Scientific, Loughborough, UK). Images were taken using Zeiss AxioImager with Apotome fluorescence microscope. Negative controls were included to avoid the formation of false-positive signals by incubating the samples without the primary antibody overnight. Neutrophil numbers in the central and corneal periphery were manually calculated from the stained images from at least four sections of the same sample.

### 4.12. Statistical Analysis

Statistical analysis was performed using GraphPad Prism v6.0. Data are expressed as mean of values of at least three separate biological donors with two technical repeats ± SEM. One-way or two-way repeated-measures ANOVA and Tukey’s multiple comparisons tests were used for statistical comparisons, while two single variables were measured with two-tailed *t* test. *p*-values were considered significant (*/$, *p* < 0.05; **/$$, *p* < 0.01; ***/$$$, *p* < 0.001; ****/$$$$, *p* < 0.0001).

## Figures and Tables

**Figure 1 ijms-21-05849-f001:**
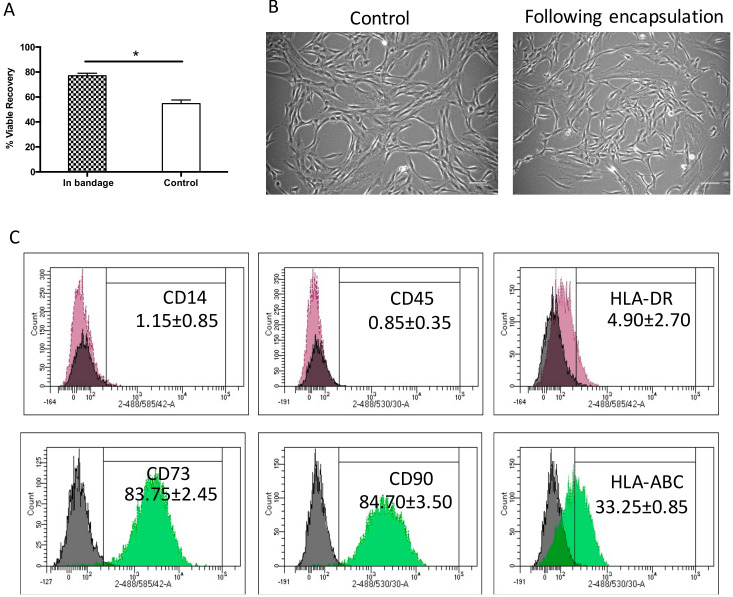
Hypothermic storage effect on human adipose-derived stem cell bandages. (**A**) Adipose-derived mesenchymal stromal cells’ (Ad-MSCs’) viable recovery was tested following 72 h storage at 15 °C comparing cells encased in bandages (in bandage) and non-encapsulated controls. (**B**) Ad-MSCs were cultured upon release from encapsulation at day 1 compared with normally grown controls. (**C**) Immunophenotype markers following Ad-MSCs released from bandages. Values are presented as a mean ± SEM from three separate donors. * represents significance, (*p* < 0.05). Scale bar = 100 µm. Abbreviations: Control, non-encapsulated; In bandage, encapsulated; HLA, human leukocyte antigen.

**Figure 2 ijms-21-05849-f002:**
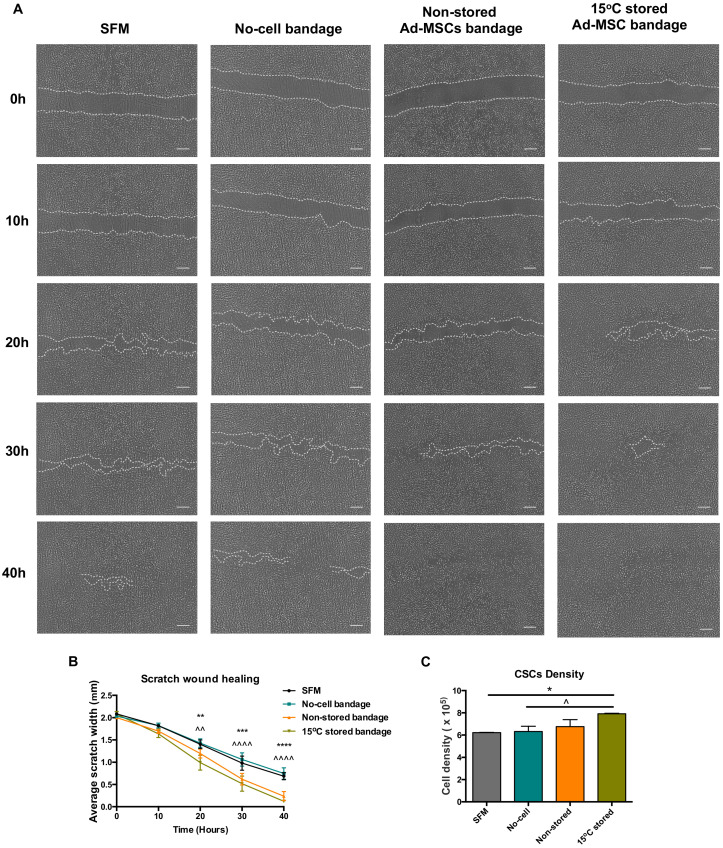
Wound closure analysis. (**A**) Representative images of scratch wound closure at times 0, 20, and 40 h taken using Iprasense holographic microscope, (**B**) average scratch width, and (**C**) corneal stromal cells (CSC) cell density following wound closure taken using Alamar blue assay of non-stored. Scale bar = 300 µm. Values are presented as mean ± SEM from three separate corneal stromal cells donors, and asterisks represent significance between SFM and 15 °C stored Ad-MSCs bandages, whereas symbols represent significance between no-cell bandages and 15 °C-stored Ad-MSCs bandages (****/^^^^, *p* < 0.0001; ***, *p* < 0.001; **/^^, *p* < 0.01; */^, *p* <0.05). SFM, Serum-free media; no-cell bandages, alginate bandages formed without Ad-MSCs; non-stored bandage, bandages used immediately following formation; and 15 °C stored bandages, bandages used following 72 h storage.

**Figure 3 ijms-21-05849-f003:**
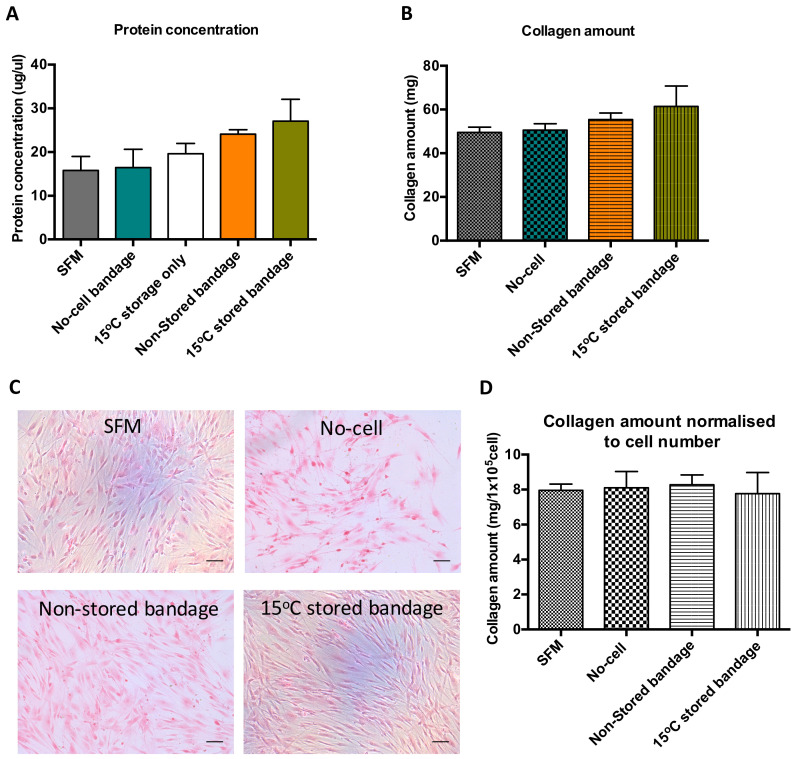
Ad-MSCs bandages stored at 15 °C exhibited increase in soluble protein concentration and collagen amount. (**A**) Assessment of soluble protein concentration, (**B**) collagen amount, (**C**) representative images of collagen stained CSCs cultures, and (**D**) collagen amount normalised to CSC number. SFM, Serum-free media; no-cell bandages, alginate bandages formed without Ad-MSCs; non-stored bandage, bandages used immediately following formation; and 15 °C stored bandages, bandages used following 72 h storage. Values are presented as mean (SEM) of three separate corneal stromal cells donors.

**Figure 4 ijms-21-05849-f004:**
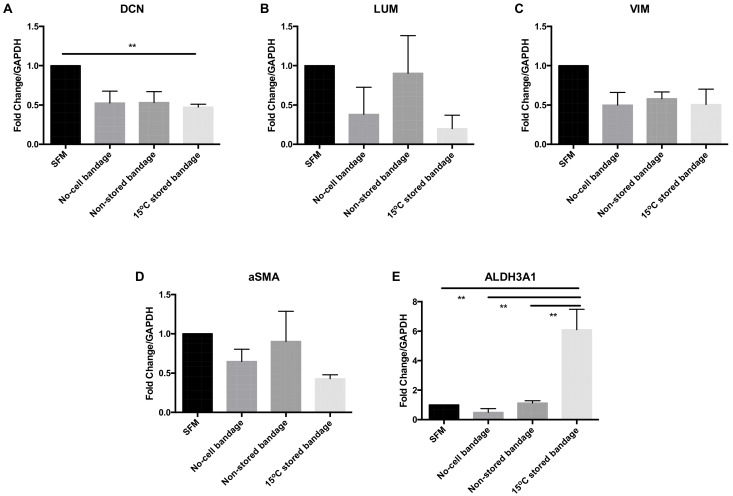
Corneal stromal cells marker expression at the gene transcription level. (**A**) Decorin (*DCN*), (**B**) lumican (*LUM*), (**C**) vimentin (*VIM*), (**D**) αSMA, and (**E**) ALDH3A1 expressions were assessed. SFM, Serum-free media; no-cell bandages, alginate bandages formed without Ad-MSCs; non-stored bandage, bandages used immediately following formation; and 15 °C-stored bandages, bandages used following 72 h storage. Values are presented as mean ± SEM of three separate corneal stromal cells donors, asterisks represent significance between conditions; ** = *p* < 0.01.

**Figure 5 ijms-21-05849-f005:**
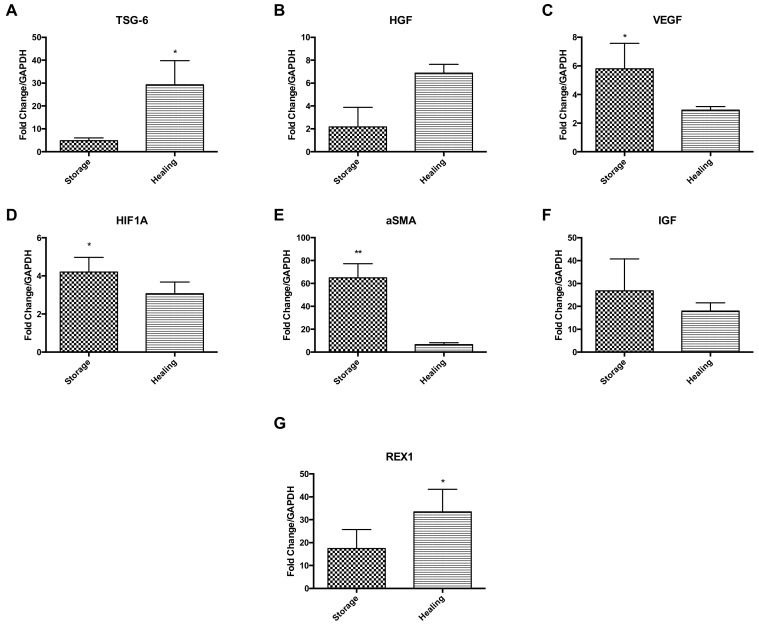
Key Ad-MSCs markers expressions were assessed, comparing stored bandages to bandages used in healing. (**A**) TSG-6, (**B**) HGF, (**C**) VEGF, (**D**) HIF1A, (**E**) αSMA, (**F**) IGF, and (**G**) REX1. Storage, Ad-MSCs bandages stored at 15 °C; Healing; Ad-MSCs bandages stored at 15 °C and used in scratch healing. Values are presented as mean ± SEM of three separate adipose-derived stem cells donors; asterisks represent significance: ** = *p* < 0.01; * = *p* < 0.05.

**Figure 6 ijms-21-05849-f006:**
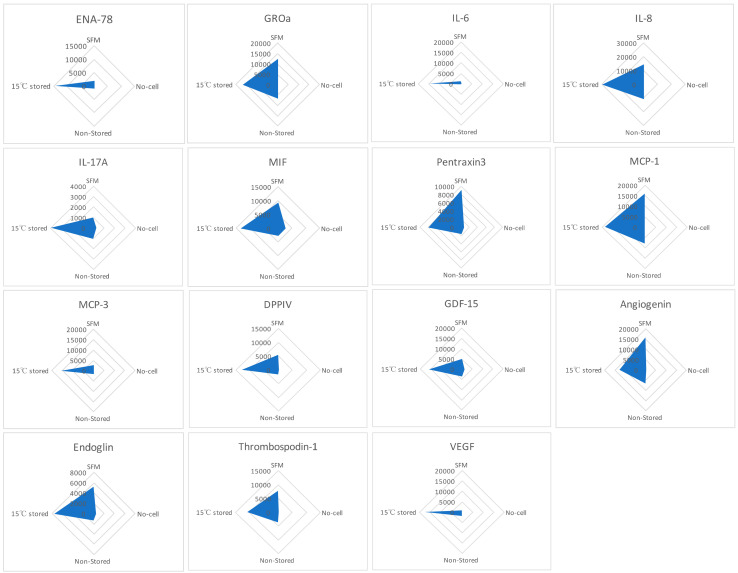
Cell production of paracrine factors. Human XL Cytokine Array kit was used to measure the relative expression of paracrine factors produced each condition. Graphs show the relative expression of epithelial-neutrophil-activating peptide (ENA-78), growth-regulated alpha protein (GROa), interleukin (IL-6, IL-8, IL-17A), microphage migration inhibitory factor (MIF), pentraxin3, monocyte chemoattractant protein (MCP-1, MCP-3), dipeptidyl peptidase-4 (DPPIV), growth/differentiation factor 15 (GDF15), angiogenin, endoglin, thrombospondin1, and vascular endothelial growth factor (VEGF). SFM, Serum-free media; no-cell bandages, alginate bandages formed without Ad-MSCs; non-stored bandage, bandages used immediately following formation; and 15 °C-stored bandages, bandages used following 72 h storage. Values are presented from one biological donor and two technical repeats.

**Figure 7 ijms-21-05849-f007:**
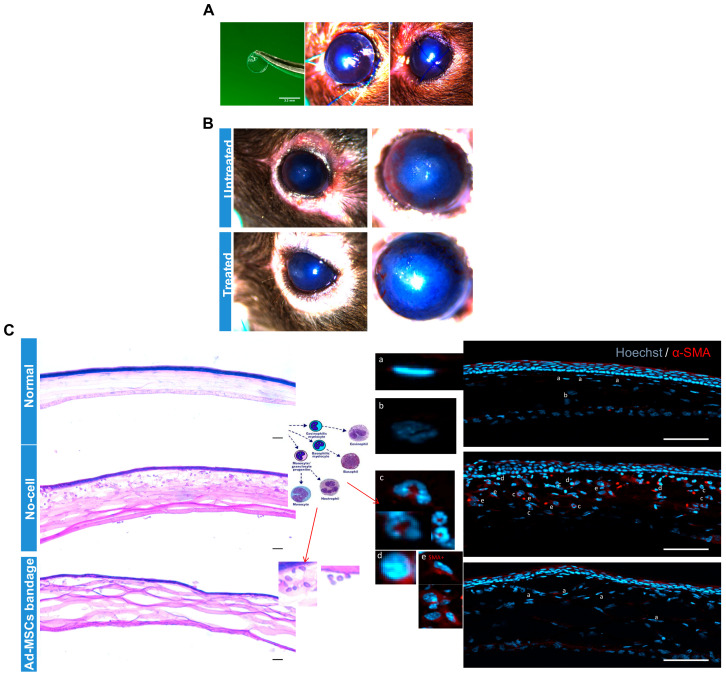
In vivo study: Epithelial debrided mouse corneas treated with alginate bandages with or without adipose-derived mesenchymal stem cells (Ad-MSCs). Ad-MSCs were encapsulated in calcium alginate gels (approx. 1 mm thickness) at a density of 20 × 10^6^ cells/mL and stored for 3 days at 15 °C. (**A**) Following epithelial debridement injury to the cornea, 3.5 mm diameter gels either without cells (untreated) or with cells (treated) were punched out and applied directly to the wounded cornea, and the eyelids were sutured closed. (**B**) Gels were left in situ for 4 days, following which the suture and gel were removed and the eye evaluated. (**C**) Eyes were imaged at 7 days post-injury before being enucleated and processed for histological analysis. H&E staining and immunofluorescence showed low levels of immune cells in the normal and Ad-MSC bandage-treated injured corneas but high levels of monocytes and neutrophils (Cc and Cd) in the untreated corneas. αSMA expression by keratocytes was absent in the normal and Ad-MSC bandage treated corneas (C and Ca) but present in the untreated corneas (C and Ce). Scale bar = 50 μm. αSMA: alpha smooth muscle actin.

**Figure 8 ijms-21-05849-f008:**
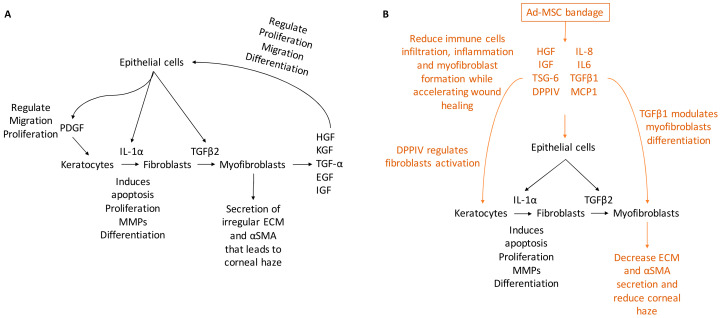
A schematic representing corneal wound healing. (**A**) Upon injury, a cascade of growth factors and cytokines released by the epithelial and stromal cells that plays an essential role in wound healing, such as IL-1α, PDGF, HGF, KGF, TGF-α, EGF, and IGF. IL-1α and PDGF are produced by the epithelial cells to mediate stromal cell response by inducing cells at the edge of the wound to undergo apoptosis while others proliferating secret MMPs and transdifferentiating into fibroblast. Epithelial cells also produce TGFβ2 that regulate cells transdifferentiating to myofibroblasts that secret irregular ECM and αSMA, which leads to corneal haze. (**B**) Ad-MSCs are able to produce paracrine factors from within the bandage that accelerate wound healing and reduce inflammation, such as IL-6, IL-8, and MCP-1 that regulate inflammatory response; DPPIV regulates fibroblasts activation and TGFβ1 that modulate myofibroblasts differentiation. Thus, the addition of Ad-MSCs reduces corneal haze and improves healing.

**Table 1 ijms-21-05849-t001:** Description of primers used in real-time PCR for corneal stromal keratocytes and adipose-derived stem cells gene expression analysis.

Gene	Primers 5′ → 3′
Decorin (DCN)	F: CTGCTTGCACAAGTTTCCTG R: GACCACTCGAAGATGGCATT
Vimentin (VIM)	F: CCTCCTACCGCAGGATGTT R: CTGTAGGTGCGGGTGGAC
a-smooth muscle actin (αSMA)	F: CTGAGCGTGGCTATTCCTTC R: TTCTCAAGGGAGGATGAGGA
Aldehyde dehydrogenase 3 (ALDH3A1)	F: CCCCTTCAACCTCACCATCC R: GTTCTCACTCAGCTCCGAGG
Lumican (LUM)	F: CCTGGTTGAGCTGGATCTGT R: TAGGATGGCCCCAGGA
GAPDH (housekeeping gene)	F: AGCCGAGCCACATCGCTGAG R: TGACCAGGCGCCCAATACGAC
Hepatocyte growth factor (HGF)	F: GTGAATACTGCAGACCAATGT R: CCAGAGGCATTGTTTTCTTGC
Insulin-like growth factor-1 (IGF-1)	F: GCTGGTGGATGCTCTTCAGT R: TTGAGGGGTGGGCAATACAT
Hypoxia-inducible factor-1A (HIF1A)	F: CCAGAAGAACTTTTAGGCCGC R: TGTCCTGTGGTGACTTGTCC
Vascular endothelial growth factor (VEGF)	F: AGGAGGGCAGAATCATCACG R: CCAGGGTCTCGATTGGATGG
Reduced expression 1 (REX1)	F: CCTCATTCATGGTCCCCGAG R: CACCCTTCAAAAGTACACCG
Tumour necrosis factor-inducible gene 6 protein (TSG-6)	F: AAGGATGGGGATTCAAGGAT R: TTTTTCTGGCTGCCTCTAGC

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
