# Peer review of "Hypothermically Stored Adipose-Derived Mesenchymal Stromal Cell Alginate Bandages Facilitate Use of Paracrine Molecules for Corneal Wound Healing"

_ijms, 2020, doi:10.3390/ijms21165849_

Round 1

Reviewer 1 Report

1. Lot of mistakes and inappropriate sentences. Explanation of results should be improved significantly. Uniformity should be maintained in explaining the results. p-values are shown to some graphs only.

2. Under topical and general anesthesia, the left cornea and limbus were treated with 1M sodium hydroxide.
Too high concentration of NaOH used for in vivo experiments (please check literature reports)

3. Ad-MSCs viability within the alginate bandages were tested following 72-hour storage at 15°C in SFM where there was x22.3 (SD 1.1)-fold increase in viable cell recovery compared to non-alginate encapsulated (no bandage) (p = 0.0463).

22.3 fold increase??? There is no such huge difference observed between 2 conditions.
And we also see less number of algenate encapsulated cells in the given area following recovery from storage compared to the normally grown cultures.

Fig.1A results need to be explained properly in detail.

4. Please explain the term non-alginate encapsulated (no bandage) viability. What is the meaning of the viable recovery of without bandage cells in this figure 1A. Does it represents the group of Ad-MSCs just kept at 15 degrees
for 72 hours without encapsulation? If yes, these cells also show a recovery rate of above 50% following 72 hours treatment at 15 degrees and needs to be shown for comparision in Fig 1B.
What are the control cells and how the control cells were treated in this experiment? This also needs to be explained in the methods and results sections.

5. Controls were also missing in Fig 1C. What is the menaing of Ad-MSC bandages containing human Ad-MSCs +/- in the Fig 1 legend?

6. In Fig 2, no significant differences in wound closure is observed under all experimental conditions.
There is a clear vast difference in the initial scratch areas of SFM, no bandage compared to the Ad-MSC bandages.
So, the observed differences in the results of scratch width represented the same behavior pattern and are not convincing.

7. Explain the condition/group of '15 degrees storage only' in the results.
In Fig.3C, highest number of cells present in Ad-MSCs bandages stored at 15°C and this high number also could be the reason for the increased soluble protein concentration. This is not explained in results and also in discussion.

8. Uniformity should be maintained in graphical representation of various experimental conditions in Fig.4.
From the results shown in Fig.4, it seems that, the decreased expression of DCN, LUM, VIM, alpha SMA, ALDH3A1 in CSCs,
in all bandage groups irresptive of the presnece of Ad-MSCs or not might be due to the effect of the material used for the encapsulation rather than the presence of Ad-MSCs.

9. In explaining the results of Fig.5, the authors mentioned Fig 8 (the actual number of figures are only 7 in the manuscript).

10. No differences can be seen in the Fig. 7A and B as the authors claim in results.

11. In results, authors claim that no differences in collagen production were found in all conditions (Fig. 3). However, in discussion they again contradict their claim.

12. Discussion needs a lot of improvement and should clearly discuss the results obtained.
The authors failed to explain the reasons of enhanced paracrine factor release under hypothermic conditions instead they
discuss about hypoxia. If hypoxia is the reason for their claims, they need to prove the existing hypoxic conditions in their experimental settings.
Why the collagen amount remained same in all conditions when paracrine production is enhanced in one experimental condition?

13. Incompletely cited references are observed and they should be cited correctly.

Reviewer 2 Report

The authors describe the benefitial effect of adipose-derived MSCs (AD-MSCs) alginate bandages on corneal healing after complete corneal epithelial debridement.

Furthermore, beside the experiments in vivo the authors described in vitro the favorable effect of AD-MSCs alginate bandages on the release of paracrine factors following MSCs hypothermic storage. AD-MSCs in alginate bandages maintained the viability during cell hypothermic storage.

The study is important for clinical practice. Corneal wound healing is very often associated with increased corneal hydration and strong proinflammatory cytokines induction leading to the excessive inflammation and corneal healing with vascularization and the loss of corneal transparency.  

AD-MSCs withing alginate bandages seem to be helpful in corneal healing processes.   

I recommend the acceptance of the study for printing in the International Journal of Molecular Sciences. 

Author Response

Thank you for your comments. 

Reviewer 3 Report

In this study, the authors showed that the viability of AD-MSCs were enhanced under the hypothermic storage via encapsulating them within alginate gel. Moreover, authors investigated that the improved effect of using Ad-MSCs bandages in healing corneal stromal keratocytes in vitro and in vivo chemical wound. I think that readers in field (study) based on corneal wound healing have an interest, although, there are some (spelling) error, and the sample number was not mentioned. Authors need to address the points of concern listed below.

My comments were shown below.

1) Why dose the author select the temperature (15℃). Is this temperature optimal? Have you ever compared the results of experiments with other temperatures? Please mention the rationale for setting the temperature to 15℃.

2) Author showed the changes in gene expression of corneal stromal cell and key Ad-MSCs markers. Does the author confirm that protein expression is also up?

3) Figure 5: Please mention the cell viability of Ad-MSCs in the gels after the experiment.

4) Author showed that the therapeutic effect of AD-MSCs on corneal wound healing. Please show the mechanism as Scheme.

5) The sample number is not shown. Please the number in the Fig. legend.

6) There are some spelling errors.

・® → Please revise to superscript

・in vitro, in vivo→ Please revise to italic

・via → Please revise to italic

Line 213: please revise “Institute”

Line 220: please change to secs rom seconds

Figure 1B, 3C: there are garbled

Figure 3A, B, D: Please collect to “Non-stored bandage” from “Non-Stored bandage”.

Line 297, 337: Please collect to “15℃・・・” from “15. ℃・・・”.

Round 2

Reviewer 1 Report

  1. Actually this is unbelievable that the authors now claim that they used a different epithelial debridement method instead of 1M NaOH for their in vivo studies. This clearly shows how much care the authors took during presenting their findings to the scientific community while writing this manuscript.
  2. What is the exact meaning of “The images are representative images of three separate biological donors and 2 technical repeats”……this explanation is still not convincing and the figure as well. Please include the images for the time points 10h and 30h also in the figure as represented in the graph.
  3. Uniformity should be maintained in graphical representation of various experimental conditions in Fig.4.

Its not about the axis……meaning the representation of the bar graphs should be the same in all experimental conditions for all the genes analysed…..For example, no cell bandage was depicted differently in all the gene analysis. Instead stick to one design of the bar graph in all gene expression analysis……please change this.

Author Response

Corrections done as requested. 

Reviewer 3 Report

The authors answered all questions. On the other hand, please revise following:

  1. Please add space between value and mins. For example. 2 mins
  2. Fig. 2, 3, 4, 6 legends: please add space "following" and "72 hours"

I recommend the publication of the manuscript after the minor revision.

Author Response

Corrections done as requested. 

Round 3

Reviewer 1 Report

Thanks for correcting the manuscript as requested.